# Extrinsic Bayesian Optimization on Manifolds

**Yihao Fang** [1], **Mu Niu** [2], **Pokman Cheung** [3] **and Lizhen Lin** [1,*]

1   Department of Applied and Computational Mathematics and Statistics, University of Notre Dame,
    Notre Dame, IN 46556, USA
2   School of Mathematics and Statistics, University of Glasgow, Glasgow G12 8QQ, UK
3   London, UK
*   Correspondence: lizhen.lin@nd.edu

**Abstract:** We propose an extrinsic Bayesian optimization (eBO) framework for general optimization problems on manifolds. Bayesian optimization algorithms build a surrogate of the objective function by employing Gaussian processes and utilizing the uncertainty in that surrogate by deriving an acquisition function. This acquisition function represents the probability of improvement based on the kernel of the Gaussian process, which guides the search in the optimization process. The critical challenge for designing Bayesian optimization algorithms on manifolds lies in the difficulty of constructing valid covariance kernels for Gaussian processes on general manifolds. Our approach is to employ extrinsic Gaussian processes by first embedding the manifold onto some higher dimensional Euclidean space via equivariant embeddings and then constructing a valid covariance kernel on the image manifold after the embedding. This leads to efficient and scalable algorithms for optimization over complex manifolds. Simulation study and real data analyses are carried out to demonstrate the utilities of our eBO framework by applying the eBO to various optimization problems over manifolds such as the sphere, the Grassmannian, and the manifold of positive definite matrices.

**Keywords:** Bayesian optimization; optimizations on manifolds; embedding; extrinsic gaussian process

## 1. Introduction

Optimization concerns best decision-making, which is present in almost aspects of society. Formally speaking, it aims to optimize some criterion, called the *objective function*, over some parameters of variables of interest. In many cases, the variable to optimize possesses certain constraints that should be incorporated or respected in the optimization process. There are studies on constrained optimization incorporating linear and nonlinear constraints, including Lagrange-based algorithms and interior points methods. Our work focuses on an important class of optimization problems with *geometric constraints* in which the parameters or variables to be optimized are assumed to lie on some manifolds, a well-characterized object in differential geometry. In other words, we deal with *optimization problems on manifolds*. Optimization on manifolds has abundant applications in modern data science. This is motivated by the systematic collection of modern complex data that take the manifold form. For example, one may encounter data in the forms of *positive definite matrices* [1], shape objects [2], subspaces [3,4], networks and orthonormal frames [5]. Statistical inference and learning of such data sets often involve optimization problems over manifolds. One of the notable examples is the estimation of the Fréchet mean for statistical inference, which can be cast as an optimization of the Fréchet function over manifolds [6–8]. In addition to various examples with data and parameters lying on manifolds, many learning problems in big data analysis with the primary goal of extracting some lower-dimensions structure, and this lower-dimensional structure is often assumed to be a manifold. Learning this lower-dimensional structure often requires solving optimization problems over manifolds such as the Grassmannian.

The critical challenge for solving *optimization problems over manifolds* lies in how to appropriately incorporate the underlying geometry of manifolds for optimization. Although there has been a fast development in optimization (over Euclidean spaces in general), and it is an extremely active ongoing research area, there is a tremendous challenge for extending theories and algorithms developed in optimization over Euclidean spaces to manifolds. The optimization approach on manifolds is superior to performing free Euclidean optimization and projecting the parameters back onto the search space after each iteration, such as in the projected gradient descent method. It has been shown to outperform standard algorithms for many problems [9–21]. Some of those algorithms, such as the Newton method [22,23], conjugate gradient descent algorithm [3], steepest descent [24], and trusted region method [25–28], have recently been extended to the Riemannian manifold from the Euclidean space, and most of the methods require the knowledge of the gradients [29–33]. However, in many cases, analytical or simple forms of gradient information are unavailable. In other cases, the evaluations and calculations of the gradient or the Hessian (second form for the case of the manifold) can be expensive. There is a lack of gradient-free methods for optimization problems when the gradient information is not available or expensive to obtain or the objective function is expensive to evaluate. In these scenarios, gradient-free methods can be appealing alternatives. Many gradient-free methods are proposed for optimization problems in the Euclidean space, especially the state-of-the-art Bayesian optimization method, which has emerged as a very powerful tool in machine learning for tuning both learning parameters and hyperparameters [34]. Such algorithms outperform many other global optimization algorithms [35]. Bayesian optimization originated with the work of Kushner and Mockus [36,37]. It received considerably more attention after Jones' work on the Efficient Global Optimization (EGO) algorithm [38]. Following that work, innovations developed in the same literature include multi-fidelity optimization [39], multi-objective optimization [40], and a study of convergence rates [41]. The observation by Snoek [34] that Bayesian optimization is useful for training deep neural networks sparked significant interest in machine learning. Within this trend, Bayesian optimization has also been used to choose laboratory experiments in materials and drug design [42], in the calibration of environmental models [43], and in reinforcement learning [44].

However, most studies mainly focus on the input domain in a Euclidean space. In recent years, with the surging collection of complex data, it is not common for the input domain to have such a simple form. For instance, the inputs might be restricted to a non-Euclidean manifold. Ref. [45] proposed a general extrinsic framework for GP modeling on manifolds, which depends on the embedding of the manifold in the Euclidean space and the construction of extrinsic kernels for GPs on their images. Ref. [46] learned a nested-manifold embedding and a representation of the objective function in the latent space using Bayesian optimization in low-dimensional latent spaces. Ref. [47] exploited the geometry of non-Euclidean parameter spaces arising in robotics by using Bayesian optimization to properly measure similarities in the parameter space through the Gaussian process. Recently, Ref. [48] implemented Riemannian Matern kernels on manifolds to place the Gaussian process prior and applied these kernels on the geometry-aware Bayesian optimization for a variety of robotic applications, including orientation control, manipulability optimization, and motion planning.

Motivated by the success of Bayesian optimization algorithms, in this work, we propose to develop the eBO framework on manifolds. In particular, we employ an extrinsic class of Gaussian processes on manifolds as the surrogate model for the objective function and utilize the uncertainty for calculating the acquisition function. An acquisition function can also be defined from this surrogate to decide where to sample [49]. The algorithms are demonstrated and applied to both simulated and real data sets to various optimization problems over different manifolds, including spheres, positive definite matrices, and the Grassmannian.

We organize our work as follows. In Section 2, we provide a streamlined introduction to Bayesian optimization on manifolds employing the extrinsic Gaussian process (eGP)

on manifolds. In Section 3, we present a concrete illustration of our eBO algorithm in the extrinsic setting in terms of optimizing the acquisition function from eGP. In Section 4, we demonstrate numerical experiments on optimization problems over different manifolds, including spheres, positive definite matrices, and the Grassmann manifold, and showcase the performance of our approach.

## 2. Bayesian Optimization (BO) on Manifolds

Let $f(x)$ be an objective function defined over some manifold $M$. We are interested in solving

$$\mu = \mathrm{argmin}_{x \in M} f(x). \tag{1}$$

Typically, the function $f$ lacks a known special structure such as concavity or linearity that would make it easy to optimize using techniques that leverage such structure to improve efficiency. Furthermore, we assume one only needs to be able to evaluate $f(x)$ without having to know first or second-order information, such as gradient, when evaluating $f$. We refer to problems having this property as "derivative-free". Lastly, we seek to know if the global optimizer exists. The principal ideas behind Bayesian optimization are to build a probabilistic model for the objective function by imposing a *Gaussian process prior*, and this probabilistic model will be used to guide to where in $M$ the function is next evaluated. The posterior predictive distribution will then be computed. Instead of optimizing the usually expensive objective function, a cheap proxy function is often optimized, which will determine the next point to evaluate. One of the inherent difficulties lies in constructing *valid Gaussian processes on manifolds* that will be utilized for Bayesian optimization on manifolds.

To be more specific, as above, let $f(x)$ be the objective function on $M$ for which we omit the potential dependence on some data; for now, the goal is to find the minimum: $\mu = \arg\min_{x \in M} f(x)$. Let $f(x) \sim GP(\nu(\cdot), R(\cdot, \cdot))$, $x \in M$, be a Gaussian process (GP) on the manifold $M$ with mean function $\nu(x)$ and covariance kernel $R(\cdot, \cdot)$. Then we evaluate $f(x)$ at a finite number of points on the manifold following a multivariate Gaussian distribution, that is,

$$(f(x_1), \ldots, f(x_n)) \sim N((\nu(x_1), \ldots, \nu(x_n)), \Sigma),$$
$$\Sigma_{ij} = cov(f(x_i), f(x_j)) = R(x_i, x_j).$$

Here $R(\cdot, \cdot) : M \times M \to \mathbb{R}$ is a covariance kernel defined on the manifold, which is a *positive semi-definite kernel* on $M$. It states that, for any sequence $(a_1, \ldots, a_n) \in \mathbb{R}^n$, $\sum_{i=1}^{n} \sum_{j=1}^{n} a_i a_j R(x_i, x_j) \geq 0$.

After some evaluations, the GP gives us closed-form marginal means and variances. We denote the predictive means and variance as $\nu(x, \mathcal{D})$ and $\sigma^2(x, \mathcal{D})$, where $\mathcal{D} = \{x_1, \ldots, x_n\}$. The acquisition function, which we denote by $a : M \to \mathbb{R}^+$, determines which point in $M$ should be evaluated next via a proxy optimization

$$x_{next} = \arg \max_{x \in M} a(x). \tag{2}$$

There are several popular choices of acquisition functions that are proposed for Bayesian optimization in Euclidean space [49]. The analogous version can be generalized to manifolds. Under the Gaussian process prior, these functions depend on the model solely through its predictive mean function $\nu(x, \mathcal{D})$ and variance $\sigma^2(x, \mathcal{D})$. In the preceding, we denote the best current value as

$$x_{best} = \mathrm{argmin}_{x_n} f(x_n). \tag{3}$$

We denote the cumulative distribution function of the standard normal as $\Phi(\cdot)$. We adopt one of the acquisition functions as

$$a(x) = \Phi(r(x)), \text{ where } r(x) = \frac{f(x_{best}) - \nu(x, \mathcal{D})}{\sigma(x, \mathcal{D})}, \tag{4}$$

which represents the *probability of improvement*. Other acquisition functions, such as knowledge gradient and entropy search on manifolds, could also be developed, but here we focus on probability improvement. As seen above, a key component of Bayesian optimization methods on manifolds is utilizing Gaussian processes on a manifold $M$, which are non-trivial to construct. The critical challenge for constructing the Gaussian process is constructing *valid covariance kernel* on manifolds. In the following sections, we will utilize extrinsic Gaussian processes on manifolds by employing extrinsic covariance kernels via embeddings [45] on the specific manifold.

## 3. Extrinsic Bayesian Optimization (eBO) on Manifolds

In this section, we will present the extrinsic Bayesian optimization algorithm on manifolds with the help of the illustrated eGP on manifolds. The key component of the extrinsic framework is the embedding map. Let $J : M \to \mathbb{R}^D$ be an embedding of $M$ into some higher dimensional Euclidean space $\mathbb{R}^D$ ($D \geq d$) and denote the image of the embedding as $\tilde{M} = J(M)$. By definition of an embedding, $J$ is a smooth map such that its differential $dJ : T_x M \to T_{J(x)} \mathbb{R}^D$ at each point $x \in M$ is an injective map from its tangent space $T_x M$ to $T_{J(x)} \mathbb{R}^D$, and $J$ is a homeomorphism between $M$ and its image $\tilde{M}$. Figure 1 provides a visual illustration of equivariant embedding. See Section 4 in [6] for some concrete examples of equivariant embeddings for well-known manifolds, such as the space of the sphere, symmetric positive definite matrices, and planar shapes. As mentioned above, we use eGP as the prior distribution of the objective function, and the main algorithm is described in Algorithm 1. Let $f(x) \sim eGP(m(\cdot), R(\cdot, \cdot))$ with prior mean $m(x)$. $R(\cdot, \cdot)$ is an covariance kernel on $M$ induced by the embedding $J$ as:

$$R(x, z) = \tilde{R}(J(x), J(z)), \tag{5}$$

where $\tilde{R}(\cdot, \cdot)$ is a valid covariance kernel on $\mathbb{R}^D$.

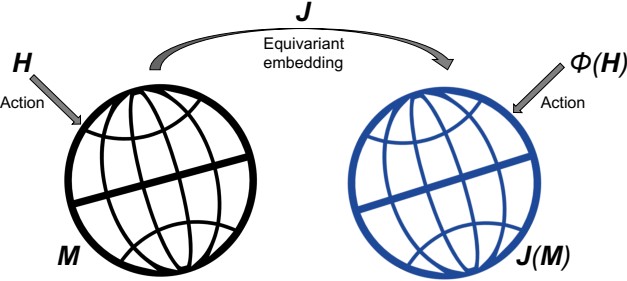

**Figure 1.** A simple illustration of equivariant embeddings.

---

**Algorithm 1** Extrinsic Bayesian optimization on manifolds.

---

Initialize $x_1, \ldots, x_k \in \tilde{M}$ ($k \geq 1$);
Let $x_{best}^0 = \text{argmin}\{f(x_1), \ldots, f(x_k)\}$ and $\mathcal{D} = \{x_1, \ldots, x_k\}$;
**for** $s = 0, 1, \ldots, T - 1$ **do**
    $x_{next} = J^{-1}(\arg\max \tilde{a}(\tilde{x}))$;
    $x_{best}^{s+1} = \text{argmin}\{(f(x_1), \ldots, f(x_k), f(x_{next})\}$;
    Update $\mathcal{D} = \{\mathcal{D}, x_{next}\}$
Return $x_{best}^T$

---

As above, let the predictive means and variance be $v(x, \mathcal{D})$ and $\sigma^2(x, \mathcal{D})$, where $\mathcal{D}$ denotes the data. The acquisition function defined by Equation (4) represents the probability of improvement. To find out the next point to evaluate, we need to maximize the acquisition question. Since we use the zero mean eGP where $m(\cdot)$ equals 0, the expression of $v(x, \mathcal{D})$ and $\sigma(x, \mathcal{D})$ are given as follows:

$$v(x, \mathcal{D}) = K(x_D, x)^T (K(x_D, x_D) + \sigma_n^2 I)^{-1} y, \tag{6}$$

$$\sigma(x, \mathcal{D}) = K(x, x) - k(x_D, x)^T (K(x_D, x_D) + \sigma_n^2 I)^{-1} k(x_D, x). \tag{7}$$

Note that $a(x)$ depends on $x$ through $J(x)$, let $a(x) = \tilde{a}(\tilde{x})$ where $\tilde{x} = J(x) \in \tilde{M}$. Therefore, the optimization of $a(x)$ over $x$ is *equivalent to the optimization of $\tilde{a}(\tilde{x})$ over $\tilde{M}$*. Let $\tilde{x}^*$ be the optimizer of $\tilde{a}(x)$ over $\tilde{M}$, i.e.,

$$\tilde{x}^* = \text{argmin}_{\tilde{x} \in \tilde{M}} \, \tilde{a}(\tilde{x}). \tag{8}$$

Then, one has

$$x^* = J^{-1}(\tilde{x}^*), \tag{9}$$

where

$$x^* = \text{argmin}_{x \in M} \, a(x). \tag{10}$$

The key is to solve (8). We consider the gradient descent or Newton's method over the submanifolds $\tilde{M}$ since the gradient is easy to obtain on $\tilde{M}$. Let $grad \, \tilde{a}(\tilde{x})$ be the gradient of $\tilde{a}(\tilde{x})$ in the Euclidean space, then the gradient of $\tilde{a}(\tilde{x})$ over the submanifold $\tilde{M}$ is given by

$$\mathcal{P} grad \, \tilde{a}(\tilde{x}), \tag{11}$$

where $\mathcal{P}$ is the projection map $P$ from $T_{\tilde{x}} \mathbb{R}^D$ onto the tangent space $T_{\tilde{x}} \tilde{M}$. We propose the following gradient descent Algorithm 2 for finding $\tilde{x}^*$ of $\tilde{a}(\tilde{x})$ over $\tilde{M}$.

---

**Algorithm 2** Gradient algorithms for optimization $\tilde{a}(\tilde{x})$ along the submanifold $\tilde{M}$.

---

**for** $t = 0, 1, \ldots, T$ **do**
    Let $\tilde{x}_0 \in \tilde{M}$ be an initial point.
    $\tilde{x}_{t+1} = \exp_{\tilde{x}_t} \lambda \mathcal{P} grad \, \tilde{a}(\tilde{x}) \mid_{\tilde{x}_t}$,
where exp is the exponential map on the submanifold. This exponential map is
with respect to the metric by restricting the Euclidean metric onto the image
manifold or submanifold.

---

## 4. Examples and Applications

To illustrate the broad applications of our eBO framework, we utilize a large class of examples with data domains on different manifolds, including spheres, positive definite matrices, and the Grassmann manifold. We construct the extrinsic kernels for eGPs based on the corresponding embedding map on the specific manifold from [45]. In this section, we show the simulation studies in Sections 4.1 and 4.2 and the main application on real data in Section 4.3. In Section 4.1, the Fréchet mean estimation is carried out with data distributed on the sphere. In Section 4.2, the eBO method is applied to the matrix approximation problem on Grassmannians. Lastly, Section 4.3 considers a kernel regression problem on the manifold of positive definite matrices, which has essential applications in neuroimaging. We use the eBO method to solve the regression problem and show the difference between healthy samples and HIV+ samples. For all examples in this section, we consider the squared exponential kernel from [45] as follows:

$$K_{ext}(x,z) = \alpha exp(-\beta \rho(x,z)^2), \tag{12}$$

where $\rho(x,z) = \|J(x) - J(z)\|$. The explicit form of embedding $J$ depends on the specific manifolds, which would be illustrated in the simulation cases. We initially set kernel coefficients as $\alpha = 1, \beta = 0.5$, and the regularization coefficient $\sigma = 0.01$. After adding a new point to the data in Algorithm 2, we also update those coefficients by maximizing the likelihood of all data.

### 4.1. Estimation of Fréchet Means

We will first apply the eBO method to the estimation of sample Fréchet means on the sphere. Let $x_1, \dots, x_n$ be $n$ points on the manifold, such as the sphere. The sample Fréchet function is defined as

$$f_n(x) = \frac{1}{n} \sum_{i=1}^{n} \rho^2(x, x_i), \tag{13}$$

and the minimization of $f_n(x)$ leads to the estimation of sample Fréchet mean $\mu_n$, where

$$\mu_n = \arg\min_{x \in M} \frac{1}{n} \sum_{i=1}^{n} \rho^2(x, x_i). \tag{14}$$

The evaluation of $f_n(x)$ at the data point $x_1, \dots, x_n$ will be our data points for guiding the optimization to find the Fréchet mean. The choice of distance $\rho$ leads to different notions of means on manifolds. In the extrinsic setting, we can define an extrinsic mean via some embedding of the manifold onto some Euclidean space. Specifically, let $J : M \to \mathbb{R}^D$ be some embedding of the manifold onto some higher-dimensional Euclidean space $\mathbb{R}^D$. Then one can define the extrinsic distance as

$$\rho(x,z) = \|J(x) - J(z)\|, \tag{15}$$

where $\| \cdot \|$ is the Euclidean distance on $\mathbb{R}^D$. This leads to the extrinsic Fréchet mean. Fortunately, the extrinsic mean has a close form expression, namely $\mu_n = \mathcal{P}_{\tilde{M}}\left(\frac{\sum_{i=1}^{n} J(x_i)}{n}\right)$, where $\mathcal{P}$ stands for the projection map onto the image $\tilde{M} = J(M)$.

In our simulation study, we estimate the extrinsic mean on the sphere $S^2$ and evaluate the performance of our eBO method by comparing it to the gradient descent (GD) algorithm in terms of convergence to the ground truth (the true minimizer given above). The embedding map $J$ is the identity map $I : \mathbb{S}^2 \to \mathbb{R}^3$. As shown in Figure 2, we simulate some data on the sphere, and the goal is to find the extrinsic mean by solving Equation (14). Specifically, those $x_i$ ($i = 1, \dots, n$) points in black are sampled on the circle of latitude, which leads to the south pole $(0, 0, -1)$ as the true extrinsic Fréchet mean. In terms of our BO Algorithm 1, we first sample those blue points randomly on the sphere as initial points and initialize the covariance matrix by evaluating the covariance kernel at these points. Then, we randomly select the direction on the sphere to maximize the acquisition function based on the eGP. In each iteration, we mark the minimizer as the stepping point in Figure 2 and add it to the data for the next iteration. Not surprisingly, those stepping points in red converge to the ground truth (south pole) after a few steps. We also compare our eBO method with the gradient descent (GD) method on the sphere under the same initialization. As illustrated in Figure 3, although our BO method converges slightly slower than GD at the first four steps, the BO method achieves better accuracy with a few more steps. It confirms the quick convergence and high accuracy as the advantages of the eBO method.

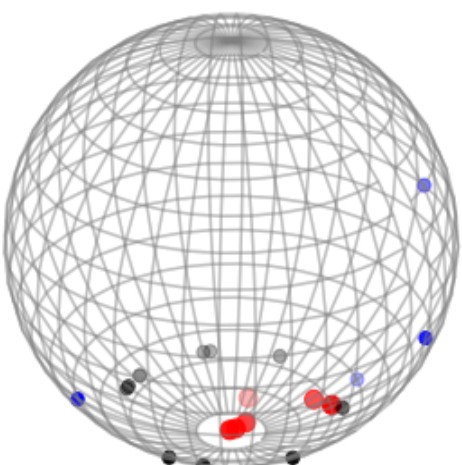

**Figure 2.** The iteration steps of the eBO method on the sphere $S^2$. The data points $x_1, ..., x_n$ are plotted as those black points on the same latitude of the sphere. The true extrinsic mean is the south pole $(0, 0, -1)$. We start with those random blue points on the sphere. The outputs of iterations in our Algorithm 1 are marked as red points on the sphere, converging to the ground truth.

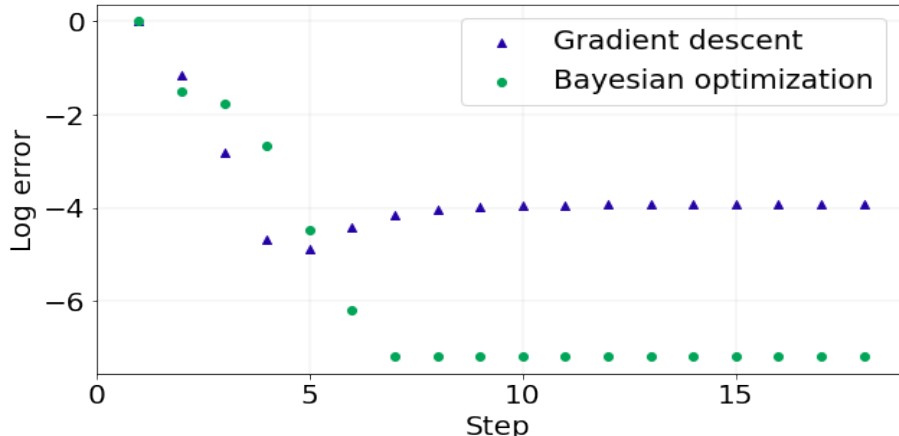

**Figure 3.** We compare the eBO method with the gradient descent (GD) method on the sphere by calculating the log error, the L2 distance from the true extrinsic mean. Although the eBO method converges slower than the GD method in the first four steps, by adding those stepping points into the eGP, it achieves better numerical precision with more steps.

### 4.2. Invariant Subspace Approximation on Grassmann Manifolds

We investigate two important manifolds, the Stiefel manifolds and the Grassmann manifolds (Grassmannians). The Stiefel manifold is the collection of $p$ orthonormal frames in $\mathbb{R}^n$, that is, $Stiefel(p, n) = \{X \in \mathbb{R}^{n*p} : X^T X = I_p\}$. Moreover, the Grassmann manifold is the space of all the subspaces of a fixed dimension $p$ whose basis elements are $p$ orthonormal unit vectors in $\mathbb{R}^n$, which is $Grass(p, n) = \{span(X) : X \in \mathbb{R}^{n*p}, X^T X = I_p\}$. Those two manifolds are closely related. The key difference between a point on the Grassmannian and a point on the Stiefel manifold is that the ordering of the $p$ orthonormal vectors in $\mathbb{R}^n$

does not matter for the former. In other words, the Grassmannian could be viewed as the quotient space of the Stiefel manifold modulo $O(p)$, the $p * p$ orthogonal group. That is, $Grass(p, n) = Stiefel(p, n)/O(p)$.

We consider the matrix approximation problem on the Grassmannian subspace manifold and apply the eBO method to solve it. Given a full rank matrix $F \in \mathbb{R}^{n*m}$, without loss of generality, we assume $n <= m$ and $rank(F) = n$, the goal is to approximate this matrix $F$ in the subspace $Grass(p, n)$ with $p < m$. From any matrix $X \in Grass(p, n)$, we approximate the original matrix $F$ from Algorithm 3.

---

**Algorithm 3** Matrix approximation on the Grassmann manifold.

---

Initialize the orthogonal matrix $X \in Grass(p, n)$;
Initialize the matrix $W \in \mathbb{R}^{p*m}$ and calculate the column $W_j \in \mathbb{R}^{r*1}$ below;
**for** $j = 1, \ldots, m$ **do**
　$W_j = (X^T X)^{-1} X^T F_j$, where $F_j$ is the $j$th column of $F$;
Return $XW$ as the approximation.

---

Here we consider the approximation error in Frobenius norm where $W$ depends on $X$:

$$X = \arg \min_{X \in Grass(p,n)} \|XW - F\|_F, \tag{16}$$

where the objective function $f = \|XW - F\|_F$. In our simulation, we consider the matrix $F \in \mathbb{R}^{3*6}$ with $rank(F) = 3$ and $p = 2$. As we know, this matrix approximation problem achieves its minimal when $X \in span(U_1, U_2)$ where $U$ is the left matrix in the Singular Value Decomposition (SVD) and $U_j$ denotes its $j$th column. Let $\hat{U} = [U_1, U_2]$ contain the first two columns of the SVD part $U$. To apply the eBO method, we first sample some matrix following $X_i = \hat{U} + \frac{i*(-1)^i}{2}$, $i = 1, ..., 6$ as initial points. Similar to the sphere case, we have to map the minimizer of the acquisition function back to the manifold via the inverse embedding $J^{-1}$. In $Grass(2, 3)$, for a given matrix $X \in \mathbb{R}^{3*2}$, $J^{-1}$ indicates us to apply the SVD decomposition to $X^T X$, then keep the first two columns. Since the closed form of the loss gradient is untouchable, we compare the BO method with the Nelder–Mead method. As shown in Figure 4, our eBO method converges to the ground truth in a few steps faster than the Nelder–Mead method.

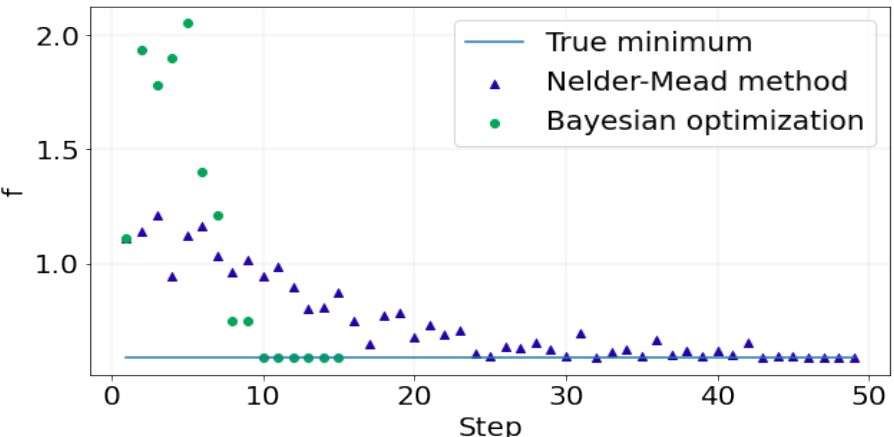

**Figure 4.** We compare the eBO method with the Nelder–Mead method on the matrix approximation problem on the Grassmannian. Since the optimal solution is the whole subspace, we calculated the value of function $f$ instead of the error. The minimum of f is around 0.5578 and is plotted as the blue line in the figure. It cannot achieve zero loss due to the low dimension constraint. The Nelder–Mead method achieves the minimal subspace around 25 steps and becomes stable after 40 steps. On the other hand, the eBO method converges to the minimum after 10 steps, much faster than the Nelder–Mead method, showing the quick convergence to the optimal solution.

### 4.3. Positive Definite Matrices

Lastly, we apply the eBO method to a regression problem with the response on positive definite matrices. Let $(z_1, y_1), \ldots, (z_n, y_n)$ be $n$ points from a regression model in which $z_i \in \mathbb{R}^d$ and $y \in M = SPD(p)$, where $SPD(p)$ stands for $p$ by $p$ positive semi-definite matrices. We are interested in modeling the regression map between $z$ and $y$.

Let $K(\cdot, \cdot) : \mathbb{R}^d \times \mathbb{R}^d \to \mathbb{R}$ be a kernel function defined on covariate space $\mathbb{R}^d$. For example, one can take the standard Gaussian kernel. Let $g(z) \in M$ be the regression map evaluated at a covariate level $z$. We propose to estimate $g$ as

$$g(z) = \arg\min_{y \in M} \sum_{i=1}^{n} \frac{1}{h} K_h(z, z_i) \rho^2(y, y_i), \tag{17}$$

where $h$ is the bandwidth parameter of the kernel function $K$. We take $\rho$ to be the extrinsic distance on the $SPD(p)$ manifold. For the given input or covariate $z$, we denote $f_n(y) = \sum_{i=1}^{n} \frac{1}{h} K_h(z, z_i) \rho^2(y, y_i))$ as the objective function and look for the minimizer $g(z)$ as the weighed Fréchet mean on the $SPD(p)$ manifold. In order to calculate the extrinsic distance on the $SPD(p)$ manifold, we choose the embedding $J$ as the log-map on the matrix, which can map the $SPD(p)$ into $p$ by $p$ real symmetric matrices $Sym(3)$:

$$\log : SPD(p) \to Sym(p). \tag{18}$$

For example, let $A \in SPD(p)$ with a spectral decomposition $A = U\Lambda U^{-1}$; we have the log-map of $A$ as $\log(A) = U \log(\Lambda) U^{-1}$ where $\log(\Lambda)$ denotes the diagonal matrix whose diagonal entries are the logarithms of the diagonal entries of $\Lambda$. Moreover, embedding $J$ is a diffeomorphism, equivariant with respect to the actions of $GL(p, \mathbb{R})$, the $p$ by $p$ general linear group. That is, for $h \in GL(p, \mathbb{R})$, we have $\log(hAh^{-1}) = h \log(A) h^{-1}$. On the other hand, the inverse of embedding $J^{-1}$ is the exp-map on the matrix, which takes the exponential function on the diagonal entries under the spectral decomposition. Then we obtain the extrinsic distance of two matrices $A_1, A_2 \in SPD(p)$:

$$\rho(A_1, A_2) = \| \log(A_1) - \log(A_2) \|, \tag{19}$$

where $\| \cdot \|$ denotes the Frobenius norm of the matrix (i.e. $\|A\| = (Tr(AA^T)^{1/2}))$.

In our analysis, we focus on the case of $p = 3$ and $d = 1$, which have important applications in diffusion tensor imaging (DTI), designed to measure the diffusion of water molecules in the brain. In more detail, diffusion represents the direction along the white matter tracks or fibers, corresponding to structural connections between brain regions along where brain activity and communications happen. DTI data are collected routinely in human brain studies. People are interested in using DTI to build predictive models of cognitive traits and neuropsychiatric disorders [45]. The diffusions are characterized in terms of diffusion matrices, represented by 3 by 3 positive definite matrices. The data set consists of 46 subjects with 28 $HIV+$ subjects and 18 healthy controls. Those 3 by 3 diffusion matrices were extracted along one atlas fiber tract of the splenium of the corpus callosum. All the DTI data were registered in the same atlas space based on arc lengths, with 75 tensors obtained along the fiber tract of each subject, which has been studied in [45] in a regression and kernel regression setting. We aim to show the difference between the positive (HIV+) and control samples based on kernel regression results.

We consider the arc length tensor as the covariant $z \in \mathbb{R}$ and the diffusion matrix as $y \in SPD(3)$ in our framework at each arc length $z$. In more detail, the arc length variable $z$ ranges from 0 to 48 with 75 different locations. We consider $K(\cdot, \cdot)$ as the standard Gaussian kernel with bandwidth $h = (4 * \hat{\sigma}_z^5 / (3 * 75))^{0.2} = 6.33$ following the rule of thumb of the univariate variable. Overall, the dimension of the whole dataset is $46 * 75 * 3 * 3$, including 28 $HIV+$ samples and 18 healthy samples. For each sample, the dimension is $75 * 3 * 3$ consisting of the matrix from $SPD(3)$ over all 75 locations. Moreover, we separately apply

the kernel regression model (17) to the $HIV+$ group and the healthy group, yielding the corresponding estimators $g_1(z)$ and $g_2(z)$ for those two groups. That is, for each location $z_i$, we consider the objective function $f_n(y)$ with data $y_i$ from those two groups separately. With the help of our eBO method, we obtain $g_1(z_i), g_2(z_i) \in SPD(3)$ from the $HIV+$ group and the healthy group, respectively. As shown in Figure 5, we observe the difference between $g_1(z_1)$ and $g_2(z_1)$, the estimated diffusion tensors at the first arc length $z_1$ from the $HIV+$ group and the healthy group, in particular for those diagonal entries. Furthermore, we carry out the two-sample test [7] between $g_1(z_1), ..., g_1(z_{75})$ from the $HIV+$ group and $g_1(z_1), ..., g_1(z_{75})$ from the healthy group and yield an extremely small $p$-value equal to $O(10^{-5})$. This result indicates the difference between those two groups.

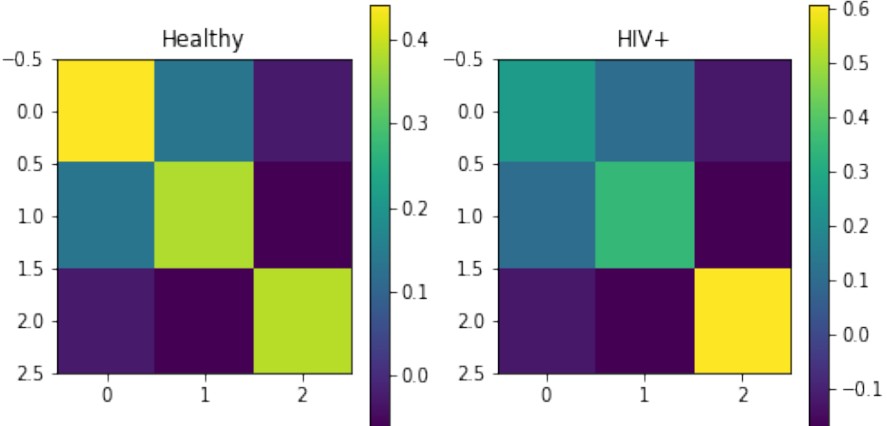

**Figure 5.** Estimated diffusion tensors $g_1(z_1), g_2(z_1)$ at first arc length from the healthy group and the HIV+ group are shown as $3 * 3$ matrix. The horizontal and vertical axis denotes the rows and columns of the matrix. Entry values inside the matrix are represented in different colors. Based on the color bar, differences between healthy and HIV+ groups could be observed, especially on the diagonal elements.

## 5. Discussion and Conclusions

We propose a general extrinsic framework for Bayesian optimizations on manifolds. These algorithms are based on the Gaussian process's acquisition function on manifolds. Applications are presented by applying the eBO method to various optimization and regression problems with data on different manifolds, including spheres, Stiefel/Grassmann manifolds, and the spaces of positive definite matrices. As a gradient-free approach, the eBO method shows advantages compared to the gradient descent method and Nelder–Mead method in our simulation study. In future work, we will investigate the intrinsic Bayesian optimizations on manifolds based on the intrinsic Gaussian processes such as the ones based on heat kernels [50].

**Author Contributions:** Conceptualization, L.L. and M.N.; methodology, L.L., M.N. and P.C.; software, M.N. and Y.F.; validation, Y.F.; writing—original draft preparation, L.L.; writing—review and editing, L.L., M.N. and Y.F.; All authors have read and agreed to the published version of the manuscript.

**Funding:** We acknowledge the generous support of NSF grants DMS CAREER 1654579 and DMS 2113642. M. N acknowledges support for this paper from EPSRC grants EP/W021595/1 and EP/X5257161/1.

**Data Availability Statement:** Not applicable.

**Conflicts of Interest:** The authors declare no conflict of interest.

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
