# Peer review of "Extrinsic Bayesian Optimization on Manifolds"

_algorithms, doi:10.3390/a16020117_

Round 1

Reviewer 1 Report

The paper proposes an extrinsic Bayesian optimization (eBO) method for optimization on a manifold. The eBO method relies on an embedding map and its inverse, a covariance kernel in the ambient space, and an acquisition function based on probability of improvement. The empirical performance of eBO is justified through three examples.

The paper is written clearly and easy to follow. The main merit of the new eBO method is that the algorithm is simple and straightforward for practical implementation under a variety of different manifold setups, as demonstrated in the numerical examples. Therefore, I consider eBO as a useful contribution. 

Some minor comments:

1. It is necessary to provide more details on which the covariance kernels have been used in the three examples in Section 4, including the choice of hyperparameters and \sigma_n^2 in equations (6) and (7).

2. For the example in Section 4.3, maybe the authors need to first demonstrate that eBO can indeed find the optimizer of the regression map through a simulation study before applying it to the real data. Also the value of bandwidth h and the distance \rho have not been specified. The embedding map J and its inverse for this example are a bit abstract, and I wonder if the authors can give the math formulas to calculate J and its inverse.

Author Response

We would like to thank the reviewer for the valuable comments. 

Reviewer 2 Report

In this work, the authors propose a novel approach (Extrinsic BO) for optimization on manifolds. They employ extrinsic GPs as probabilistic models, first by embedding the manifolds onto higher dimensional Euclidean spaces and then, defining a valid kernel on image manifolds. The authors solved some optimization problems over manifolds and showed the efficiency gains in comparison to some known methods.

Comments:

1- Intro: " In particular, we employ an extrinsic class of Gaussian processes on manifolds as the surrogate model for the objective function and quantify the uncertainty in that surrogate." In BO, the uncertainty of the probabilistic model is used to calculate a utility function (acquisition function) to decide where to sample next. But, quantifying the uncertainty is a terminology that does not describe such process. It actually refers to uncertainty quantification and recognizing sources of uncertainty. Perhaps a re-wording is needed here.

2- Extrinsic Bayesian Optimization on manifolds, definition of an embedding: A graphical example can be helpful to the readers to understand the concept behind embedding map. At least, citing some references for more related discussions is suggested.

3- Algorithm 1: it is repeating explanations from bullet points above it. Bullet points may be removed and extra explanations comes after the algorithm if needed.

4- Examples and Applications:

part 1: Authors compare their proposed method performance (eBO) with GD. A more valid comparison would be running computations multiple times with different starting points. Then, figure 2 then should show the average performance with confidence intervals. The same thing applies for part 2 and figure 3.

Line 154: "minimize the acquisition function". In BO, the goal can be minimizing a function (as is in this example) and the next best experiment is selected by maximizing an acquisition function. Please address this accordingly to avoid any confusions.

Figure2: What does loss show in figure 2? if it is showing the difference between true minimizer and best minimum found, it should be stated that the loss refers to the error. This is more confusing in figure 3. In the caption, it is mentioned that the minimum is 0.5578. Is this the loss or the function value? It seems to be the function values in figure 3. Please clarify this in the manuscript and figure axes names.

Authors claim the results in figure 2 show the convergence, however, only 6 steps are completed. More number of iterations are needed to make any conclusion about convergence (in addition to multiple simulations).

part 3: the results in figure 4 need more explanations. Interpreting the results are not trivial. Authors should discuss the results more. Is this the main application the authors try to implement eBO on? If it is a test like parts 1 and 2, is there any performance comparison with any other methods to show the advantage of eBO approach? This should be clarified in the manuscript.

5- typos: Page 5, on top: "Let grad  ̃a( ̃x) in the Euclidean
space, then the gradient ...". The sentence needs to be edited.

Author Response

(The authors gave the same response as above.)

Round 2

Reviewer 1 Report

The authors have sufficiently addressed the comments in my previous review.

There are two small typos in the revision.

1. On page 4 Section 3 line 8, "See Section [9]..." seems to miss the section number, since the reference [9] is a book.

2. On page 6 equation (12), the kernel misses a negative sign inside the exponent.

Author Response

Thank you so much for your comments which we have addressed in the revision. 

Reviewer 2 Report

Thank you for revising the manuscript.

Author Response

Thank you so much for your effort and time in reviewing our paper.  

Round 3

Reviewer 1 Report

The authors have corrected the typos. I am happy to accept the current manuscript.